# Should We Want to Be Loved Unconditionally and Forever?

**Troy Jollimore**

Department of Philosophy, California State University, Chico, CA 95929-0730, USA; tjollimore@csuchico.edu

**Abstract:** People often say that romantic love should be unconditional, and they often want romantic love to last forever. These claims and desires are presumably linked: part of the reason it would be good for love to be unconditional is that it is assumed that such love, being detached from changing conditions, would last forever. This article argues that there are, indeed, kinds of unconditional and permanent love that are worth wanting, but also kinds that are not, and attempts to clarify just what it is that is valuable about these kinds of romantic love.

**Keywords:** commitment; Christopher Cordner; enduring love; Susan Mendus; romantic love; Kieran Setiya; unconditional love

## 1. Introduction

People often say that romantic love should be unconditional, and they often want such love to last forever. Or, if not forever—people do die, after all—then at least, for the rest of their lives. (These claims, to be clear, are made about various forms of love, but I will be focusing on romantic love in particular.) These claims and desires are presumably linked: part of the reason it would be good for love to be unconditional is that such love, being detached from changing conditions and thus somewhat independent of the shifting sands of circumstance, is more likely to be permanent. Or so, at any rate, it is often assumed. I want to explore the question of whether these kinds of love—unconditional love, and permanent love—are indeed worth praising and wanting, and if so, in what forms and under what conditions.

## 2. The Right Kind of Permanence

I will begin with the idea of permanent love. It is not uncommon for people to promise to love their lovers forever; and many people want to be the object of such promises. But can it possibly be reasonable to make such a promise? One might worry that a promise to do anything for the rest of one's life is an over-extension of our powers of reasonable commitment. Who knows what my circumstances will be, or even what I will be like, forty years from now? Moreover, love, whether or not it *is* an emotion, certainly involves feelings and emotions in a central way, and this raises many issues with respect to commitments and promises. In general, I do not have direct voluntary control over my emotions. A cutting remark might anger me, whether I want to get angry or not; I am somewhat afraid of needles, even though I know the fear is not rational; and I am not able just to start loving you—or, if I do love you, to *stop* loving you—at will. However, ordinary promises, at any rate, tend to concern actions and outcomes that are under our control.

Emotions, by their nature, change with time: they correspond to and reflect current circumstances in one's life, and alter in accordance with alterations in their objects. (It makes sense to feel fear when faced with a threat, but the fear fades once the threat has passed.) Emotions are, by nature, evaluative, and whether I continue to feel the positive evaluative feelings I currently experience toward a given person will depend both on how I change over time (I might come to be inclined to appreciate different values as I age and change) and how that person changes (she might lose the qualities that ground my current positive responses). A promise to love somebody forever might make sense if it

were possible somehow to freeze us both in time, putting an end to all change. However, this, of course, is neither possible nor desirable.

The Romantic poet Percy Bysshe Shelley is making a related point when he draws a comparison between romantic love and belief:

> Constancy has nothing virtuous in itself . . . Love is free; to promise for ever to love the same woman is not less absurd than to promise to believe the same creed; such a vow in both cases excludes us from all enquiry. The language of the votarist is this: the woman I now love may be infinitely inferior to many others; the creed I now profess may be a mass of errors and absurdities; but I exclude myself from all future information as to the amiability of the one and the truth of the other, resolving blindly, and in spite of conviction, to adhere to them. ([1], pp. 45–48)[1]

Notice that the claim, regarding beliefs, is not that we should never have stable, long-lasting beliefs. Rather, the claim sets itself against having beliefs that remain stable, and endure, for the wrong reasons. My belief that *modus ponens* is a valid argument form, for instance, or my belief that strong arguments generally provide reasons for accepting their conclusions, are certain to remain stable, not because I have formed a commitment to hold them no matter what else I come to think, but because the prospect of my coming to think something that would require that I suspend or abandon those beliefs is radically remote. Similarly, it is good when a romantic relationship endures, so long as it endures for the right reasons—reasons having more to do with the relationship itself than with my commitment to the relationship. Healthy and flourishing relationships tend to endure; while this is true for a variety of reasons, the most obvious one is surely that the question of how long a relationship endures is largely up to the participants, and in general the healthier and stronger a relationship is, the more the participants will *want* it to endure. (There is no quick account to be had of what it is for a relationship to be healthy and strong, but certain generalities, though perhaps defeasible, may be safely asserted: ordinarily a good relationship promotes the well-being and happiness of the participants, helps them pursue and choose to pursue projects and ends that bear genuine value, and encourages them to grow, for instance.) The problem is that it is possible to want a romantic relationship to endure, and decide that it will, for the wrong reasons: to demonstrate the strength of one's steadfast commitment, to prove wrong those skeptics who greeted news of the connection by predicting it would never last, to live up to a traditional ideal according to which such relationships must be maintained in all circumstances and regardless of the cost, and so on. Such motivations, rather than being turned outward toward the person one loves, are often turned inward: the lover might desire to be, or appear to be, a certain sort of person, one who never breaks off his commitments, one whose will is admirably strong. Here, the appearance of loving commitment masks a kind of narcissistic self-concern. Or else, the lover's attention might simply be focused on the presumed value of commitment for commitment's sake, perhaps to the point of being willing to sacrifice not only his own well-being but his beloved's as well, in order to preserve the relationship at all costs[2]. In such cases, Shelley might say, the constancy of the lovers has, if anything, a negative value.

Consider an analogy. I know some people who are what we might call *committed* readers: having begun a book, they will read it to the end. Other people are *conditional* readers: they will read a book as long as they continue to find it rewarding—or perhaps, as long as they continue to believe that it will ultimately prove to be rewarding—but if they get to a point where they decide that this condition will probably not be met in this case, they put it aside and look for another. If I am looking for books that are worth *my* time, I will be much more interested in the choices of conditional readers than in those of committed readers. A committed reader, after all, will finish *anything*; if they say, of a given book, "I couldn't put it down," that tells me very little—they can't put *anything* down. Just so in love: here, too, some people are bound to "finish what they started," no matter how unpleasant or unrewarding it may be, whereas others are ready to end a relationship when a certain threshold has been reached. People might disagree, of course, about where that threshold is, and it is certainly as possible to be insufficiently committed

as to be excessively committed. The point, though, is that one would not say of either sort of reader that they see no value in finishing books. (Nor would one necessarily say that they are not at all committed to finishing books. The completely uncommitted person will rarely ever finish *any* book, since every book has its lulls and dry patches; a reasonable reader knows that one must get through those in order to achieve the valuable experience she seeks.) The conditional reader simply denies that there is much value in finishing a book *for its own sake*. Similarly, a conditional lover denies that there is much, if any, value in hanging in there for its own sake. However, this does not show that they do not value commitment in any way, and it certainly does not show that they do not value their relationships, or the people they have relationships with. Indeed, precisely because their focus of attention is on the other person, and not on their own determination or powers of commitment, it is plausible to think that they value their partners *better*—more directly, more sensitively, more justly—than those who would lock themselves into a permanent relationship with anyone to whom they found themselves attached, due to the excessive value they place on commitment itself.

In telling these stories in this way I have been imagining viewing these situations from the outside: what do we learn about the value of the book, or the relationship, by observing others' perseverance? A great deal, perhaps, but only if they persevere for the right reasons; or so I have suggested. However, I can also get information, in a similar way, from my own perseverance—my own inclination to finish a book rather than casting it aside, or to continue with a relationship rather than ending it. Again, however, this is true only if I persevere for the right reasons. As a result, my ability to obtain such information is dependent both on my possessing the right set of standards and practices with respect to practical reasoning—i.e., being the sort of agent who is sensitive to various sorts of experiences, rather than the sort Shelley warns us against, who is so single-mindedly committed that they will persist through an activity to the bitter end no matter how unrewarding it might become—and, along with this, enough self-knowledge to know that this is the sort of agent I am. This is no small thing: love relationships are rife with self-deception, along with deception of other flavors. There are surely many lovers who tell themselves that they are sticking with a particular relationship, and a particular lover, because that relationship or person is so wonderful, when as a matter of fact the correct explanation has more to do with their own determination and stubbornness. In general, then, the fact that a relationship endures over time is often a sign that the relationship is valuable; but this tends not to be the case when the endurance of the relationship is itself seen, by the participants, as an overriding value.

Such reflections do not challenge the idea that there may well be something good about Permanent Love, as long as it is the right sort of Permanent Love. However, they do suggest that we can acknowledge this while agreeing with Shelly that constancy has no value in itself, and thus rejecting the idea that we ought to embrace constancy for constancy's sake, or permanence for permanence's. It may still be true that we may reasonably hope that our current love turns out to be permanent, and also true that there are things we may reasonably do in order to increase the likelihood that it will—not by simply gritting our teeth and marshaling our resolve to remain with our current partner no matter what, but rather by doing the kinds of things that will cause the relationship to remain (or, if it has fallen into a sad but not irreparable state, to become) the kind of relationship that is likely to endure on its own merits. Notice that this connects the phenomenon of permanence in love with what we said earlier about the inconstancy or flexibility of emotions: both should be seen largely as indicators of values rather than as manifesting kinds of value that are themselves directly to be promoted. It feels good to be happy, but happiness ceases to function properly when we adopt ways of stimulating it that are independent of the facts and circumstances of our lives. Likewise, while anger is unpleasant to feel, it is often important that we do become angry when it is called for by the situation; the ability to turn off our anger responses would not be a boon. We should not try to stimulate or repress our emotions directly, but rather try to bring about the kinds of things that tend to spur

positive emotions and, more generally, to create the conditions, and the kind of life, in which positive emotions tend to arise. Just so, rather than forcing a relationship to be permanent, by adopting an irrevocable, circumstance-insensitive commitment, we should try to make our relationships healthy and strong in ways that will naturally promote their enduring over time.

None of this forces us to the extreme claim (which Shelley might perhaps have endorsed) that willed commitments—including such formalized commitments as marriage vows—are without value. For one thing, being dedicated to acting in the ways just suggested—ways that will tend to promote the flourishing of the relationship, thus prompting the lovers to want it to continue—is itself a kind of commitment. Moreover, an explicit commitment, particularly a public, formalized one, can be a useful bulwark against the fact that spontaneous motivations are not always reliable and sometimes fluctuate; absent such a commitment, one might be in danger of being too quick to abandon a relationship that is, in fact, inherently strong and worth preserving, as soon as one's felt desire flags[3]. *That* sort of commitment, accompanied by a promise to promote the well-being of one's partner and of the relationship, can be entirely reasonable. Thus, the fact that many lovers need to make a voluntary choice to stand by their commitments from time to time does not in itself show that their relationships are lacking. Nearly everyone goes through difficult periods now and then, and a commitment may be necessary if a relationship is to survive them.

This recognition, however, falls quite short of the kinds of exuberantly romantic utterances many lovers say, and want to hear, when making love promises. Rather, they tend to say, and to want to hear, things that sound like pop song lyrics, things such as *I will always love you* and *I'm gonna keep on loving you* and *My heart will go on*. They do not tend to say things like, *I promise that I will always try to perform the actions necessary to maintaining the quality of our relationship, which, in turn, can be expected to lead both of us to continue to desire to remain in that relationship, and hence to lead to the endurance over time of said relationship; and also that I will not break things off with you at the first sign of trouble, but will hang in there for a reasonable length of time on the assumption that the relationship is indeed worth preserving*. That is far too wordy, and too legalistic, to be romantically satisfying. Lovers, not to mention popular music lyricists, ought to be granted a certain degree of poetic license in such matters.

### 3. Unconditional Love?

Enduring relationships tend to be good relationships, and good relationships tend to endure. This simple, commonsensical statement captures a good deal of what I have said to this point, if we Combine it with the point that there are ways of trying to force relationships to endure that are too focused on decision and the will, and hence should be avoided.

Can we say the same kind of thing about Unconditional Love? One reason for thinking we cannot arises when we try to identify what is wrong with the kind of Permanent Love I have identified as not desirable. For it looks as though the answer is, quite precisely, that such love *is* unconditional: it is the unconditionality itself that is the problem. As Shelley suggests, standing by a partner (or, for that matter, an employer, a country, or a president or other leader) no matter what happens—and, in particular, no matter what that person does or becomes—is no more reasonable than continuing to hold on to a belief in the face of any and all available counter-evidence and counterargument. Permanent Love is good, we might say, not when it is achieved via an unconditional commitment, but when it is conditional by nature and survives precisely because it meets the appropriate conditions. The worry is not only that an unconditional commitment could lead us into misery and despair, tying us to a relationship we would have no way of escaping should it turn bad[4]. It is also that, by making love's commitments unconditional, we render love unable to perform many of the important roles it tends to play in our lives, and hence rob it of much of its value. To take an obvious example, the fact that we are loved ordinarily makes us feel good about ourselves, as we think it ought to; but if our partner loves us merely because

they have promised to, and hence take themselves to be obligated to do so, how could that possibly strike us as affirming[5]? Many of us would—correctly, I would argue—be dismayed, not delighted, by a love that said, in essence, "I don't love you because you are brilliant, witty, charming and beautiful; now that we are committed to one another all of those are irrelevant qualities. Indeed I would love you just as much if you were stupid, boring, morally repulsive, obnoxious, and thoroughly unpleasant to be around—as you are quite likely to become, given what the passage of time tends to do to people".

Similar reservations about unconditional love have been expressed by other philosophers. Simon Keller claims that "Being loved romantically should give us a reason to feel good about ourselves," and takes this as a reason to think that romantic love must be conditional ([7], p. 163). Russell Vannoy writes that "lovers do not want to be loved out of sheer charity . . . . They want to 'win' the heart of their lover and not merely have love handed to them because they need it" ([8], 302). Moreover, as Derek Edyvane notes, it is not necessary that you are yourself the object of another's unconditional love, in order to feel such misgivings: the knowledge that one's lover is so undiscerning that she previously loved someone who was clearly unworthy can itself be enough to diminish if not negate the value of her love for you ([9], p. 72). For his part, Neil Delaney acknowledges an apparent conflict of intuitions here, but ultimately concludes that unconditional love is not really what we want: "While on the one hand you seem to want to be loved unconditionally, at the same time you want your lover to be discerning . . . . [W]hile you seem to want it to be the case that, were you to become a schmuck, your lover would continue to love you, as would be the case if love really was unconditional, you also want it to be the case that your lover would never love a schmuck" ([10], p. 347).

There is reason, then, to worry about unconditional love; to wonder, even, whether it is really love at all.[6] However, maybe 'unconditional' does not—or, at any rate, need not—mean precisely what we have been taking it to mean. At the very least, it is worth asking whether there are ways of understanding unconditional love that might better capture the value many people do claim to see in it. Enough people see unconditional love as desirable and good that it would be risky at best, and a likely violation of the demands of charity, to conclude too swiftly that it is simply bad by nature and that there is nothing positive to be said about it. We ought to ask, then: how might love be unconditional while still retaining its value for us, and without running afoul of the kinds of concerns we have been raising? And how might such an unconditional love be related to the sort of permanent love I have already claimed is worth valuing?

The concerns I have raised about permanence suggest that we ought to ask whether a love that avoids making excessively strong claims about commitments to certain forms of future behavior (or, worse still, future feelings) can nevertheless be genuinely unconditional. Christopher Cordner has made a suggestion that is highly relevant here, proposing an account of unconditional love that is oriented more toward the present than toward the future—in particular, the distant future. Unconditionality, Cordner argues, is often better understood not "as a matter of [love's] never–never ever, come what may–failing across time," but rather as something "distinctive in the quality or character or manner of *the loving itself*." Understood in this way, the requirement that one's love be unconditional is *not* the demand that one promise or intend to stand by one's beloved forever and under any conceivable circumstances; it is, rather, the demand that the lover be *devoted* to her beloved in a sense that manifests itself in the present moment. The mere fact that a love does not end does not in itself show that it is perfect: a love that continued forever might be cold, grasping, possessive, manipulative, or deficient in quite a number of ways. Cordner claims that we should also accept the corollary: that what we really want out of love–even if we use the language of unconditionality to express our desires–is not something that would have to prove its worth by enduring forever or exhibiting "sustained continuance or repetition" under any and all circumstances, but rather a quality, or combination of qualities, that can be displayed in the present moment, something "in the loving itself" ([12], p. 5).

To claim that love is 'unconditional' in this sense, then, is not primarily to make a prediction about the future, nor to commit oneself everlastingly to a certain course of action, but rather to describe the character of the present love and the conditions under which it is being offered. From this vantage point there are indeed ways in which many instances of love can be seen to be objectionably conditional. The lover who sees love merely as an investment that promises significant returns, and who attends to his beloved only under the condition that this expectation remain in effect, would fail to love unconditionally, as would the lover who attends to her beloved only if and when it is convenient for her to do so, or the one who will "love" his beloved forever, but only as an object, a beautiful thing to display to the world. (Notice that many of these are just the sort of things that get ruled out by the language regarding "for richer or for poorer,", etc., in popular marriage vows taken or adapted from the Book of Common Prayer). Similarly, Bernard Williams' famous example of the man who saves his wife from drowning, but only after first asking himself whether his commitments to impartial morality permit him to do so, can be seen as illustrating a way in which love might be objectionably conditional: the "one thought too many" that Williams claims the husband requires himself to have—the thought that explicitly concerns the moral permissibility of saving his wife from death—makes explicit the fact that the husband takes his commitment to his wife to be conditional on its being in accordance with the demands of morality, which are, it would appear, of more fundamental significance to him ([13], p. 18).

Cordner himself, in illustrating his claim, chooses a different kind of example, one that involves not romantic love but parent–child love:

> I coached my son's community football team when he was 10. My assistant coach, Dennis, had a boy in the team. He also had a younger son, Toby, with pronounced physical and intellectual disabilities, who used to come to the football games with Dennis. I had seen only a little of Dennis with his sons away from the football field; I was vaguely aware that Toby needed quite a bit of his time. One day, at a big moment in an important match for the team, with Dennis and I both fully focused on the play, Toby came up to ask his father for something. The loving and wholly attentive patience with which Dennis turned and immediately responded to Toby has always remained with me, as a kind of [example of] what real attentiveness, real loving presence to one's child here and now is. ([12], p. 6)

What Cordner means to describe is not an ordinary moment of attentive interaction: the sort one might commonly observe, say, between a dedicated teacher and their student, which would express their professionalism and concern but not necessarily any sort of *love*. Such an interaction would not have had the same impact on Cordner, and would not serve as an example of unconditional love. I take it that we are meant to imagine that Dennis is prepared to respond to Toby in this manner regardless of the situation, no matter what other important matters he is currently occupied with and how deeply absorbed he is in them. It is precisely this, after all, that underlies the view that this is an *unconditional* love.

It is also worth saying that, though this kind of unconditional love is expressed and can be observed in actions in the present moment, we need not hold that it has *nothing* to do with endurance or other temporal considerations. Suppose, for instance, that the next day Dennis were to wake up and discover that he no longer felt any affection for Toby at all, and, as a result, he no longer treated him lovingly. This would surely serve as evidence against the claim that the incident Cordner witnessed expressed genuine unconditional love. (This is particularly so if Dennis offered an explanation of his change of emotion that was couched in conditional language of some sort; but even if Dennis were to offer no explanation at all, and had nothing to say but "The heart wants what it wants" or some similar cliché, we would take the change as a kind of evidence). The general assumption is that "unconditional" love does not fluctuate, not that it fluctuates wildly in ways that have nothing to do with conditions; and it would not be at all unreasonable, in light of this, for a person who sees that their lover's love for them is unconditional in this manner, to take this as a strong reason for thinking that it is highly likely that their lover will continue

to feel thusly about them for a considerable period of time. (Similarly, Neil Delaney, who denies that unconditional love is really what we desire, nonetheless acknowledges that romantic love involves strong commitments to one's partner ([10], p. 350)).

However, if Cordner is right, as I am inclined to think he is, then unconditional love as we ought to understand it has less to do with temporal considerations than is commonly assumed, and certainly should not be understood as a love that is insensitive to life and its changing circumstances, a love that remains fixed and rigid no matter what alters around it. We do not, then, have to be certain that Dennis's love for Toby will in fact last until one of them dies in order to say, with confidence, that his love is unconditional. The unconditionality is demonstrated in the here and now, and is entirely visible to us in this moment. It is shown by Dennis's willingness to turn all of his loving attention to Toby as soon as Toby requests or requires it—without having to check first to see whether, in light of the current situation, this is permissible, or wise, or in some way to Dennis's advantage.

## 4. Mendus on Commitments to Persons

We find something like this focus on the quality of love displayed in the present moment in Susan Mendus's commentary on marriage promises. Mendus is responding to those who are skeptical about such promises, largely for reasons we have mentioned, and in particular, due to the fact that people often change a great deal over time. While acknowledging that such changes do sometimes prompt people to alter or end their commitments to their partners, she insists that this is no reason for being skeptical either of our ability to make such commitments or of their normative force. Nor should we conclude that when making such a commitment we ought to build such conditions into it:

> [A]lthough I might give up my commitment to my husband, and give as my reason a change in his character and principles, this goes no way towards showing that only short-term promises carry any moral weight, for there is a vital distinction here: the distinction between, on the one hand, the person who promises to love and to honor but who finds that, after a time, she has lost her commitment (perhaps on account of change in her husband's character), and, on the other hand, the person who promises to love and to honor only on condition that there be no such change in character. The former person may properly be said, under certain circumstances, to have given up a commitment; the latter person was never committed in the appropriate way at all. ([14], p. 246)

By 'in the appropriate way' Mendus means to indicate the lover's being committed *to the person* they love, as opposed to endorsing or approving certain character traits the beloved may possess, or certain principles the lover may have adopted or committed themselves to:

> [A person who allows] in advance that she will love her husband only so long as he does not change in any of the aforementioned ways, fails properly to commit herself to him: for now her attitude to him seems to be one of respect or admiration, not commitment at all. Now she *does* mutter under her breath 'So long as you don't become a member of the Conservative Party'. However, the marriage promise contains no such 'escape clause'. When Mrs. Micawber staunchly declares that she will never desert Mr. Micawber, she means just that. There are no conditions, nor could there be any, for otherwise we would fail to distinguish between respect or admiration *for the principles* of another and the sort of unconditional commitment *to him* which the marriage vow involves. ([14], p. 246)

This is not unreasonable, as far as it goes. Certainly there seems to be something correct in the idea that what is central here is the issue of whether or not one is genuinely committed to the person one loves. What a great many marriage vows seem to rule out is, again, precisely those 'commitments' that misfire in as much as they are directed toward things other than the person. If I am committed to you only because you come from a wealthy family, and I want to get my hands on some of those riches, I am not committed to

*you*, and the 'for richer or for poorer' clause of many marriage vows clearly rules that out. In general, if I view you in purely instrumental terms, valuing you because of what I think you can get for me, or how I believe you can assist me in achieving my pre-established goals, and so forth, then I am not really committed to you. Properly unconditional love will, accordingly, avoid setting these sorts of conditions. Likewise, a valuable case of enduring love will be one that endures, not merely because the beloved does indeed turn out to be instrumentally valuable over the long haul in the ways the 'lover' had hoped he would be, but because the lovers are committed to each other and value each other as persons, as individuals. All of this, of course, is essentially just a recognition of a point Aristotle made about friendships in the *Nicomachean Ethics*, when he ranked character friendship above the sort of purely instrumental 'friendship' that occurs when you simply take yourself to derive benefits from your association with the friend.

However, this leaves somewhat open the question of how we are to understand what it is to be committed to a person. I do not think it will do, as Mendus seems to suggest, to say that making such a commitment implies that I simply "cannot now envisage anything happening such as would make me give up that commitment" ([14], p. 247). Similarly, we should reject her claim that valuing the principles a person lives by and the values she recognizes, and to some degree valuing that person *for* those principles and values, is not only separate from but incompatible with valuing the person herself. We should reject, that is, Mendus's view that a true lover would never be willing to admit that if the beloved were to completely abandon her principles and values, and adopt contrary ones, he would end the relationship. The first claim, about being unable to envisage certain possibilities, seems psychologically implausible. Mendus herself, after all, writes that "I might give up my commitment to my husband, and give as my reason a change in his character and principles". Perhaps there is meant to be a difference between *acknowledging* this to be true and *envisaging* it, but this is a fine line at best; at the very least more needs to be said. In my view, it seems all too easy, knowing what we know about human frailty, to envisage at least some such things; the beloved, for instance, might be seduced into joining a racist political party with an atrocious platform, or suffer some other form of radical moral deterioration. More significantly, I am quite skeptical about the strong distinction she defends between being committed to a person and valuing their principles. After all, to the extent that I identify with and approve of my own principles, values, and virtues, I might well desire that anyone who claims to love me value me for these particular aspects of my identity, rather than other things I might view as less essential or integral to who I am. I might even think that placing *some* value on these things is *necessary* in order to really love me.

Moreover, commitment to principles or values themselves can come in varying degrees, and the strength of such a commitment might very well be entirely relevant to the constancy of one's love. At the very least, an account of love and its commitments should not rule this out. Suppose that Isaac first met Isha while they were both taking part in a protest against a pipeline that was to run through a sensitive animal habitat and would contribute to climate change. Isaac was initially attracted to her, in part, because of her passionate commitment to such issues. They fell in love and were married. Over the years, however, Isaac's devotion to environmental issues faded, and he came to regard Isha's unwavering commitments as an irritant—moreover, a frequently costly and inconvenient irritant. He demanded that she give them up; when she refused, he left her. This certainly looks like a case in which Isaac's failure of commitment toward the relevant principles and virtues—the ones Isha held and instantiated, and refused to give up—was very closely tied to, and resulted in, the failure of his commitment toward *her*. Of course, in committing themselves to each other each was committing herself/himself to an individual, and not *just* to a set of principles, values, or virtues. However, there is no reason to deny that the latter sort of commitment can be *part of* committing oneself to an individual person.

One difference between committing oneself to a set of principles and values, and committing oneself to an individual person, is that the latter sort of commitment allows for certain kinds of changes, and in particular for guided change and joint growth. Many

commentators have noted that love involves a kind of openness to being guided or shaped by the beloved, so that the fact that my partner holds certain values, for instance, often causes me to endorse or adopt those same values, or to seriously consider whether I ought to do so[7]. If so, then we need not be committed to the *same* values over time in order to remain committed to our beloved; it all depends on how they change, as well. Obviously, there are no guarantees: it is possible for lovers to grow apart rather than growing together or in parallel. If this were not possible then the issue raised by Mendus simply would never arise. However, there are, after all, many relationships in which the issue does *not* arise, not because the lovers are fixed in their nature and never change, but because their mutual influence and guidance is sufficient to prevent them from changing in ways that break the relationship apart.

### 5. Qualities, Commitments, and Time

A closely related matter concerns the role of qualities. It is often thought that a genuinely unconditional love must avoid being based in any way on the qualities of the beloved, since if it were so based, this would imply that the love was not unconditional after all, but rather conditional on the person's continuing to possess those qualities. As I have argued elsewhere, the truth is surely more complex than this. ([4], pp. 135–142) To begin, we should not assume that every beloved will inevitably lose her loveable qualities; this is particularly important, perhaps, when what is in question are moral virtues, commitments to principle, and other such features. Some people do become corrupted, of course, but others manage to maintain or even improve their character over time. As far as other qualities go, the more superficial ones—physical attractiveness, agility, athleticism, and the like—do tend to peak and fade, but they are frequently replaced over time by other desirable characteristics.

The role of these newly emerged attractive properties is especially significant because, in my view—I have elsewhere referred to it as the Vision View of love—love involves a commitment to a way of seeing the beloved that centrally involves a kind of charitable attention to the beloved that includes an attitude of openness toward valuable features they might display, along with values they themselves might endorse [4]. The lover, then, is more likely than others to see and appreciate the beloved's good features, including good features that have recently arrived on the scene rather than having been there all along, or are in the process of developing. She is better positioned to do this on account of her knowledge of the beloved, based on past experience and enabled by the intimacy of their relationship; the fact that she loves this person gives her both normative reason and motivation to pursue such perceptions and to hone her ability to engage in this kind of seeing. We should remember, too, that the lover's valuing of the beloved's good features will be of a certain sort; it will *not*, for reasons already hinted at above, be merely instrumental. Human beings, after all, are not just useful, or capable of being traded for other good things, or productive of good things, and so forth. A thing that is merely *useful* is open to being appropriately traded for another thing of the same type that is equally useful, if not more so; human beings are not replaceable in this way. To borrow a bit of Kantian terminology, a human being has a dignity, not a price[8]. It is, of course, possible to 'value' human beings in such superficial ways, and to treat them accordingly. However, such valuing is not love. In as much as it is an objectifying attitude, it is indeed the antithesis of love.

Rather than being opposed to commitments, the kind of value appreciation appropriate to human beings actually involves a kind of commitment. It is not a commitment that is at all divorced from or hostile to the appreciation of value, not only because it is motivated by my appreciation of the beloved's value, but also because of its content: it is, in fact, a commitment to continuing to hold oneself open to the value of the beloved's qualities—including those new qualities that she may acquire as she changes over time—and indeed to seek them out and endeavor to learn to appreciate them even when doing so turns out to be somewhat difficult. (In other words, part of appreciating a person's value is seeing why it is worth putting a certain degree of effort into trying to continue

to learn to appreciate her value.) Moreover, the lover will for the most part not take this attitude as a matter of obligation. Rather she will be naturally and spontaneously motivated to do so. Of course, as with any extended pattern of attitude and behavior, a deliberate choice to maintain a conscious commitment may be necessary to sustain the relationship through a difficult time. For the most part, though, the commitments of romantic love are spontaneously felt and expressed, not adhered to because they are viewed as mandatory.

We might nevertheless worry that despite this, such views must still view love as conditional in an objectionable sense. After all, are there not some cases in which a person will lose all of her valuable qualities, and not develop new qualities to replace them? Kieran Setiya has put the objection as follows:

> Consider [ . . . ] the more extreme case in which I see that my reasons are gone, and I cannot find replacements. This may be rare, but as Jollimore concedes, it is not impossible. Here, the quality theorist must deny that it is rational to love. And this is a mistake. It is not true that love *must* continue in the face of radical change . . . . but [rather] that unconditional love is not irrational . . . . It is not irrational to love my wife with a constancy that would survive the loss of her admirable qualities, the things that drew me to her in the first place or that attract me now. Nor it is irrational to love one's children, come what may. ([22], p. 256)

I want to say a couple things in response to this. The first is that I am simply not sure I share Setiya's intuitions on this matter. If we really imagine the sort of "extreme example" that Setiya is describing here–one in which a person has lost *all* of her attractive qualities, and has not developed any to replace them–it is not obvious to me that in such a case, continuing to love this person *romantically*–as opposed to, say, continuing to express what Neil Delaney has called a "loving commitment," which involves care and concern but is quite another sort of thing from romantic love—could possibly be reasonable ([10], p. 350). The case of a person losing *all* of their attractive properties, and not acquiring or developing any new ones, is, at least outside of cases of complete dementia, bizarre and surely very rare. Of course, this is not to say that the person who does continue to romantically love a beloved who, say, has been in a vegetative state for months and is never coming out of it, should be criticized and told that his love is unreasonable; that might just be cruel. However, in part, this may just be a disagreement about language. I agree, after all, that there is a sense in which one may continue to love one's beloved for years after they fall into an irreversible coma or, for that matter, simply pass away. However, there would be something wrong with a person who continued to love such a beloved *in the same way* as she was loved when she was a conscious human being with whom he was actively pursuing a shared life. Whatever love remains after such an irreversible loss, it is a very different thing (Again, see [11]).

Perhaps some will be surprised to hear me say that it can be reasonable, if we take love to be in large part a response to the beloved's valuable qualities, to go on loving a person who has been dead for years. However, I see no reason to limit the qualities that ground one's love to those that are *currently* possessed by the beloved (See [4], pp. 139–41; [23]). Again, if we are speaking of objects that are valued purely instrumentally, such a limitation would be reasonable. A knife that is no longer sharp, and cannot be re-sharpened, has no value for me; it cannot cut, which is what the knife is for. In general, things that are only valuable in so far as they serve some particular function lose their value when they lose their ability to serve that function. However, people are not knives. Their values are not merely instrumental values: we do not value a person because she serves some clearly defined purpose or other. Indeed, it is a deep mistake in loving to approach one's relationship thinking only about what one can "get out of it," as if it were a resource to be plundered. The same, of course, holds of artworks, or of cultural artifacts or sites. Think of the ruins of the Acropolis. Surely our knowledge of the beauty the Acropolis *used* to possess, in the time of its full glory, is an essential part of the experiences of value it is capable of inspiring today. In all of these cases we are valuing something not as a mere

means to some further end, to which it is subordinate; we are valuing them as things in themselves. They have final rather than instrumental value.

Moreover, persons are things that exist in, move through, and develop over time in a way that objects such as knives do not. We tell stories about people, and in that way understand their value–what is worthwhile and special about them. Additionally, some of the positive attitudes we take toward people are by their very nature directed toward things that took place in the past, sometimes long ago. There is nothing strange about respecting, honoring, or admiring someone for an action or accomplishment–a display of genius, of courage and integrity, or what have you–performed many years ago. Nor is it the case that the past accomplishment only matters inasmuch as it serves as evidence that the person possesses certain traits or virtues *now* and so is likely to do something like that again at some point. Ordinarily, a person who accomplishes something significant simply remains admirable; the significance of the accomplishment for our feelings about her is not annulled after the passage of some certain span of time.

Holding that what one does when one loves is largely a matter of responding to the valuable qualities of the beloved, then, is perfectly compatible with holding that one is responding not only to current qualities but also to past qualities—particularly those the beloved possessed in previous stages of the relationship, and which were, perhaps, significant in causing the relationship to be formed in the first place. People are often loved for being beautiful, but one can also love someone for having been beautiful. Indeed, a lover is often able to see, in a way that others cannot, the vestiges of former beauty in the beloved's current face. She may well be able to see that face as genuinely beautiful, and may even be to some real degree unaware of the effects time has had on it. Accounts of love that emphasize the significance of the responsiveness of love to qualities, then, need not be opposed to those that emphasize the significance of commitment, including commitments that are in many significant respects unconditional.

Many different kinds of valuing can be involved in the appreciation of value. Even on its own, the ruins of the Acropolis surely possess a certain sort of beauty. In light of its history, and of what a historically informed observer knows about the ruins, they simultaneously embody a different kind of beauty and interest. We could imagine, as well, an individual who has a personal history that includes this site: suppose that she proposed to her husband-to-be on the spot. This would involve yet another form of value that the site would bear for her: something like the sentimental value my grandfather's pocket watch might have for me. Loving a human person tends to be complicated in just this way, with various forms of value layered on top of one another and interacting complexly: the face of my beloved is beautiful in and of itself, more so for the traces it bears of and the story it tells about her past beauty; and of course, she and I share a history, so that she represents, for me, a kind of link or bridge to that history. All of this suggests that the standard stories about meeting an even more attractive individual, and "trading up" without loss, are much too simplistic to account for the reality of what actually goes on between people[9].

## 6. Surviving Error and Immorality

Let me close by considering a different sort of worry. Setiya writes:

> Even if it works, the turn to history cannot solve a related problem: that of error. It is not irrational to love my wife with a constancy that would survive not only the loss of her admirable qualities but the discovery that they were never real. If someone can lose these qualities, there is a more peculiar case in which they were absent all along. You never acted from true kindness, always with an ulterior motive; your "jokes" were unintentional; my memories are false [ . . . ]. [I]t is not a mistake to go on loving my wife when I learn that I was wrong about her from the start. This kind of commitment conflicts with the need for valuable qualities, even past ones, as reasons for love. ([22], p. 257)

If what we are supposed to imagine is that I might somehow have been wrong about *all* of my wife's qualities—she does not, in fact, have *any* of the attractive features I thought

she had—I must confess that I am not entirely sure quite how to make sense of such a possibility. I thought her face was beautiful, but I was mistaken? I thought she was fun to be with, but really she was not? (If what I was having with her was not fun, what was it?) We can, of course, imagine that love can survive some degree of error; we all make mistakes, and I am sure each of us has a certain number of false beliefs about our friends and romantic partners. However, Setiya seems to ask us to imagine loving someone because she is kind and witty, only to discover that she is unkind and unfunny; and to discover, at the same time, that *every* positive quality I thought she had was in fact mere appearance and illusion[10]. I think, first, that I would have to be extraordinarily obtuse to have this much wool pulled over my eyes: genuine kindness is often quite apparent and fully visible in individual acts, and while an ulterior motive might be something that can be concealed on occasion, if one acts on it habitually one is sure to be found out. Beyond that, I am not sure, in fact, that I *can* imagine being *that* mistaken about somebody—unless I am in an Experience Machine or being victimized by an evil demon. Additionally, to the extent that I can imagine it, it is not at all clear why I ought to think that continuing to love her would be correct and appropriate. Indeed, the very idea of 'continuing' to love her implies that I have been loving *her*, and in this case I do not see how that could be. You cannot love someone who you do not know *at all*; just as, if your beliefs about elephants are that they are tiny insects that infest wool carpets in South American countries, you do not really have any beliefs about *elephants* at all.

Perhaps we can begin to make some sense of such a possibility by imagining that the agent's standards shift, and that he then projects those standards back in time: yes, he says, I was once callow enough to see her own callowness and impetuosity as exciting and desirable, but now I know better, now I see that she was *never* worthy of my desire! We sometimes have such thoughts, though they are rarely if ever completely true; in fact, there probably was *something* good about the qualities one now finds oneself turning against, even if, from the standpoint of what one now takes to be maturity, one finds it difficult to admit it, or to see it. The most plausible examples, perhaps, are those that focus on specifically moral qualities. For here we can, I think, imagine having been substantially mistaken about a person's value, in two distinct ways: I did not realize they had some quality; or, I knew that they had it but was not aware that the quality was bad. The latter involves, again, a shift in standards, and in this case, it does make sense to think that the current standards might reasonably be projected into the past: having realized that the person I once loved is cruel, and that it was out of my own moral inadequacy that I found her cruelty enticing, I no longer find it to be so.

Might one nonetheless go on loving, even here? "It is not a mistake," Setiya writes, "to love one's child in the face of a terrible wrongdoing, though one's love for him may be transformed" ([22], p. 258). I take it that Setiya means to include all wrongdoings, even the most terrible. Kamila Pacovská has advanced a related claim: that the best, most admirable sort of lover, a "saintly" lover, "can perceive value even in human beings with a minimal degree of qualities to be appreciated and even in those who don't appear to be human at all" ([25], p. 138). I myself find such a view neither plausible nor attractive: it seems to me there are some acts so awful—rape, genocide, torture—that they render a person unworthy of love[11]. This is particularly clear when we focus our attention on the sufferings caused by such persons and their wrongdoings: whatever the appropriate response might be, it is not love. To love the perpetrators, I think, would be to betray to victims. To demand that we love evil persons seems to involve, in Tony Milligan's nice phrase, "a problematic transfer of saintliness into the intimate domain" ([28], p. 175).

Suppose we set the extreme cases aside, and confine ourselves to cases in which the beloved's wrongdoing falls short of the truly evil. Here, I am considerably more sympathetic to the idea that to be lovingly committed to a person, and not merely to principles or virtues, must surely mean that when that person acts unvirtuously, and in ways contrary to acceptable moral principles, we do not simply abandon them, turning our backs and going off in search of someone more admirable to love. Exactly what this

means is difficult to say, and will vary considerably from case to case. One thing I do think, though, is that the love cannot remain unaltered; one cannot go in loving in the same way one did before. Thus, I am inclined to think that 'transformed' is a quite good description of what ought, and perhaps must, to go on here. However, this in itself seems to show that the love in question is not wholly unconditional. (As does the fact that we have only gotten this far by setting aside the most terrible cases).

Why does love prompt us to not to abandon (all but the worst) wrongdoers? Because love moves us to see the whole person, and so to remember that, while they have committed an offense, they are more than that, more than just *that person who did that bad thing.* That bad thing is simply one part of them, one incident in a much larger story. Each of us is far more than any particular action we might perform, and to love someone, is, in part, to recognize that we are not always at our best, and thus to place particular bad actions in the context of a whole life, including a future that potentially contains opportunities for regret, amends, and positive change. (Perhaps what sets the truly evil actions aside is that in those cases, placing the action in the context of a whole life does very little to affect our view of the person). Such a view—a loving view—shows a kind of commitment to the individual person. However, once again, such a commitment is not incompatible with commitments to principles, or virtues; nor does it preclude considering the subject's qualities. The person you love, whom you now refuse to abandon on account of your love, has displayed various virtues and admirable qualities in the past. In those moments it was possible to see their potential—perhaps only to glimpse it, if the opportunities for such displays have been few and far between. However, such glimpses, to a loving eye, can be enough. Human beings are imperfect, and undergo moments of weakness, confusion, poor judgment, and other forms of imperfection. The valuable forms of constancy that are attached to love—as opposed to the rigid, stultifying forms that Shelley rightly warns us about—are valuable, and necessary, because we are prone to such things, and thus incapable of always being our best selves.

At any rate, to say that love ought to be unconditional in this sense—or to admire real cases of it for being so—is not to assert that there are no conditions at all; the question, as always, is which conditions we should take to be relevant, what we think they demand and permit, and how much guidance they can give us in the very difficult matter of deciding what to do when someone we care about has committed a serious offense. A person who had made a truly unconditional commitment—a commitment to a relationship's being maintained, and as much as possible kept in its current form, come what may, and in this manner rendered permanent—would have to regard immoral actions (regardless of whom they harm) as irrelevant to the question of whether the relationship, or the love, ought to end. Thus, an affirmative answer to either of those questions would simply be ruled out from the start. Their opposite number, a person who is deeply averse to commitment, would be free to flee at the first sign of trouble. (I do not say that they are free to 'end the relationship,' since in the absence of commitment there could be no relationship). I am fairly confident that, as so often is the case, the ideal attitude lies somewhere between these extremes: there is some degree of commitment that represents a genuine loving attachment to another person while stopping short of rigidity or servility. I doubt that there is a general answer to the question of just how this ideal degree may be defined; if light is to be shed, it will be by exploring the details and nuances of particular cases.

I have argued that, while there are some kinds of conditions a true lover will not insist on, and some she will regard as entirely irrelevant, a literally unconditional love, one that bore no conditions whatsoever, would, if such a thing were even possible, be too rigid and alienating to be satisfying or desirable. When we say that we want to be loved unconditionally, what we really want is something else: a love that avoids certain sorts of conditions, and which manifests a genuine and strong but defeasible commitment to the persons we are, on the part of our partners. I have argued, too, that, while a love that endures over time for the right reasons is indeed a valuable thing, we should not desire a long-lasting commitment whose endurance is only guaranteed by the dogged

determination of the lover. The desire for unconditional love, or for enduring love, is not in itself a faulty or unreasonable desire, but we are apt to be misled about the nature of what would satisfy that desire, if we do not think carefully about just what it is that we are actually wishing for.

**Funding:** This research received no external funding.

**Conflicts of Interest:** The author declares no conflict of interest.

## Notes

1   Compare Elizabeth Brake's related argument, which also involves emotions: since we cannot control our emotions, and "one can't promise to do what one can't do", it follows that despite appearances, marriage vows are not and cannot be promises. See [2] However, Brake's concern is not with the responsive nature of emotions, but with the claim that we cannot control them; and her conclusion is not that we should not make such promises, but rather that they are not actually promises at all.

2   The true lover, then, will agree with Raja Halwani when he writes, "There is no point to maintaining the relationship unless doing so helps maintain the flourishing of the beloved [ . . . ] the relationship itself is not the primary good." ([3], p. 39.).

3   I discuss this and related issues in [4] (pp. 95–122).

4   Though this is a serious worry. Dan Moller, for instance, makes a great deal of it. See [5].

5   Cf. Daniel Markotivs: "[I]t is intuitively plain that if the reason why a wife remains faithful to her husband, or cares for him in a time of need, or performs any of the other myriad acts that a loving and intimate marriage involves is just that she recognizes that she is bound by her marriage vow, then her marriage is in serious trouble." ([6], p. 312).

6   Of course, as I go on to acknowledge, past qualities can also ground love. So one could hold, even on grounds that I allow as legitimate, that once you love someone, you should always love them. My own view is closer to Delaney's: while love should not fluctuate directly with (perceived) changes in the beloved's qualities, it should not remain entirely detached, either; though it might survive radical changes in the nature of the beloved (and whether it should will depend on just what those changes are), the love that emerges on the other side will not be just the same love as it was before. See Delaney's comments on "plasticity" and "responsiveness" ([10], p. 349); see also A.O. Rorty's excellent article [11].

7   Many people have proposed views that include some substantial connection between loving a person and endorsing, being guided by, or committing to that person's values, though they differ, of course, in their details. See, for instance [15–20].

8   "If [a thing] has a price, something else can be put in its place as an equivalent; if it is exalted above all price and so admits of no equivalent, then it has a dignity" [21] (p. 434).

9   An early statement of the "trading up" objection appears in [24] (p. 76).

10  To be clear, Setiya's position is not that rationality would require him to continue loving his wife in this situation. It seems to be, rather, that rationality is permissive: if he were to discover that all of his beliefs about his wife had been erroneous, rationality would not require him either to stop loving or to continue loving. See [22] (p. 257).

11  Raja Halwani and Alan Soble express similar views. Halwani: "Surely there are moral defects for which no good moral qualities can compensate, thereby blocking [a lover's] endorsement." Halwani lists racists, torturers, and supporters of ethnic cleansing among his examples, and asks, "If the lovers are not, *at the very least,* troubled by their beloveds' moral actions given their . . . avowed commitments, we would likely wonder how integrated their values are with each other and with their actions and motives" ([26], p. 227). Soble, in responding to a claim made by W. Newton-Smith that resembles Setiya's, writes: "This thesis is implausible at least in those cases in which x believes falsely that y is a saint and later discovers that y is, or becomes, a monster" ([27], p. 205).

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
