# Peer review of "Should We Want to Be Loved Unconditionally and Forever?"

_philosophies, doi:10.3390/philosophies8020034_

Round 1

Reviewer 1 Report

In this paper, the author criticizes the notion of unconditional love and suggest that a literally unconditional love that bore no conditions whatsoever, is both undesirable and impossible to imagine. In so doing, the author creates the conceptual space for thinking about unconditionality in a new way that does not make a prediction about the future, but instead describes the character of the present love and the conditions under which it is being offered.

The article makes a contribution in the ways that it is situated in the growing body of work on reason-responsive account of love by situating the loving commitments as also involving a responsiveness to reasons. For example, the author argues that not just any such reason for honoring one’s commitment to a relationship will do. This is best illustrated where the author makes use of the reader who commits to finishing a book regardless of their evaluation of its content and the reader who sets books that they start aside when they evaluate it as not being worth more of their time.

The author does not thoroughly cite work on commitment and love and one does wonder what a more thorough going analysis of promises might reveal about the nature of commitments typically found among friends and other intimate relationships. Daniel Markovits’ “Promise as an Arm’s-length Relation” is a fine example.

Overall, however, the article is well written and its arguments are clear, compelling, and decisive. I recommend this article for publication as is.

Author Response

Thank you!

I had read the Markovits before, and gave it a quick read again as a result of your prompting. I now include a quotation from it. That said, I don’t have the space here to engage with his main claims in any substantial way; and to be honest I am still making up my mind about what to think about them. I think I agree with some of what he says but not all of it; it would take more space than I have available to address it properly.

More generally, there is a good deal more to say about promises etc., of course, and I hope to address some of it in future work. I do include a bit more in the revised version of the paper (I reference Elizabeth Brake’s work, for instance), but again, limited space just doesn’t permit me to get into any of it deeply.

Reviewer 2 Report

Attached file.

Author Response

Thank you. I’ve added notes to Nozick and Kant, as suggested, and some other references as well, including some recent ones!

Reviewer 3 Report

I really liked this paper, it's nicely written and interesting. I have a few suggestions below, but nothing I'd insist on:

p. 2 on your point that it is only good if a relationship endures for reasons having more to do with the relationship itself than with commitment - does this mean that your view challenges the idea of marriage? It seems to me that part of the point of marriage is to provide an additional reason not to break up and thus to maintain one’s commitment, and that people seem to think it’s an achievement to have had a long-lasting marriage, and thus ‘been committed’, even if the marriage was not altogether happy.

p. 5 top of page – you say that Cordner says that what we really want out of love is something that can be displayed in the present moment. This seems unintuitive to me – I would have said that what we really want from love often references the future.

p. 5 the Cordner quote – this doesn’t sound like an obvious example of unconditional love to me, but rather of attentiveness. A teacher could be attentive in the same way without loving the child.

P. 7 bottom of second paragraph – I thought an example would be good here

p. 7 third paragraph – you say we need to get clear what it means to be properly committed to a person. I wondered here whether you’re using commitment as a proxy for love, or whether commitment is something distinct? It seems to me that one could be committed to another without loving them.

I thought perhaps there could have been a few more references in the paper too – e.g. Simon Keller’s ‘How do I love thee, let me count the properties’ and Derek Edyvane's 'Against Unconditional Love' seemed relevant, and perhaps also Frankfurt.

Typos:

p 3 line 4 ‘valuable’ should be ‘value’

P4 middle of second paragraph – ‘perform many the important roles’ – missing ‘of’ here

P5 middle of second paragraph - ‘the one who will “loves” his beloved’ – should be ‘love’

P7 middle of second paragraph –‘willing to admit that the beloved’ – missing ‘if’ before ‘the’

P8 bottom of third paragraph – ‘that equally useful’ – missing ‘is’ before ‘equally’

P10 second line – ‘she is serves’ – ‘is’ should be deleted

P11 fourth line of third paragraph ‘know I now’ should be ‘now I know’

P12 bottom of first paragraph – ‘of always of being’ – should be ‘of always being’
